# The Effect of Web Augmented Reality on Primary Pupils' Achievement in English

Harith A. Hussein [1,*], Majid Hamid Ali [1], Muhaned Al-Hashimi [1], Nahida Taha Majeed [2], Qabas A. Hameed [1] and Reem D. Ismael [1]

1   Department of Computer Science, College of Computer Science and Mathematics, Tikrit University, Tikrit 34001, Iraq
2   Department of English, College of Education for Humanities, Tikrit University, Tikrit 34001, Iraq
*   Correspondence: harith_abd1981@tu.edu.iq; Tel.: +964-770-200-7940

**Abstract:** The massive development of mobile computing and mobile networks attracts developers and researchers to emerge a new direction for augmented reality on the Web. Web augmented reality is inherently considered platform independent, no pre-installation is required, easy to apply, easy to access, and easy to develop. The current study aims to introduce a vision of including Web AR in school instructional material to keep the teaching methods in line with the tremendous technological growth and invest students' knowledge in this field. The main objective of this study is to develop a QR code-based tracking Web AR application to investigate the effect of Web AR on the achievement of 4th-year primary school pupils in English. The procedure of Web AR application includes two basic steps. Firstly, the convincing Web AR development platform is selected according to three evaluation criteria. Secondly, developing the Web AR with 73 English vocabularies included in the last four units of the Iraqi English pupil's book. The procedures of the study include; First, a random selection of a sample of pupils and assigning them to experimental and control groups. Second, equalize the elected pupils in the factors that may affect their performance. Then, the control and experimental groups have been taught English for 12 weeks, and finally, three achievement posttests are constructed and applied to the involved groups in order to assess their performance. Having a long period of learning on using Web AR is significant, as the pupils' tendencies and acceptance of the experiment tools require time and effort.

**Keywords:** English as second language; MindAR; mywebar; primary education; Web AR; Web-based AR

## 1. Introduction

The internet is initially designed as an independent platform to be a universal system for sharing information and services. Almost all people are familiar with internet technology; whether older or youngsters. Since the Internet dramatically contributes to student study and education. The COVID-19 pandemic has created the most significant disruption to education in modern history. The emergence of the pandemic has required educators and students across all levels of education to adapt quickly to virtual courses via the Internet. The Corona pandemic has accelerated primary school pupils' knowledge of the Internet. Therefore, this knowledge must be improved and invested in adopting e-learning as a parallel method for traditional education. In the last few years, we have witnessed Mobile Augmented Reality (MAR) applications in many areas like education, gaming, and tourism. These applications are fueled and empowered by new technologies like computer vision, machine learning, and the Internet of Things (IoT). However, MAR inherited many limitations for example hardware specifications are costly and heavy, and many MAR applications require further download and installation and lack cross-platform ability [1]. Web Augmented Reality (Web AR) or web-based augmented reality is inherently considered platform independent, no pre-installation is required, easy to apply and easy to access.

The most important instrument that civilization uses to communicate with one another, whether orally or in writing, is language. Because it is regarded as the language of science, technology, commerce, trade, and other fields, the English language is one of the most important languages in the entire world [2]. Additionally, it is now a required subject in schools and colleges all around the world so that students can learn about new cultures and ways of feeling, thinking, and doing. [3]. None native students of English do not have the ability to communicate and produce meaningful sentences in English. They often encounter certain difficulties in writing and speaking English. Learning English as a Foreign Language (EFL) is affected by many interrelated factors such as, learning environment, teachers' proficiency, teaching methods, and materials. The focus of E-learning is on learning to generate ideas and learners are encouraged to apply knowledge and make sense of the material covered in class. The teachers have to apply more practical and student-centered approaches for teaching English, especially for learners who face difficulties [4]. The teacher should enable his/her student to use English in social context effectively at various grades of instruction [5]. The Web AR implicitly supports socialization which drives information from the content within a discussion by teacher-student and student-student interaction may enable the students to comprehend English learning well. The current study could hold and focus on students' attention best. Moreover, the material of EFL pupils' textbook, lack the element of entertainment and they lack authenticity. Also, the instructional method adopted by EFL teachers for teaching dialogues or conversations is mostly traditional. By and large, those teachers do not keep pace with modern methods and strategies.

Pupils waste their time browsing the web for gaming, social media, and entertainment. The current study aims to introduce a vision that includes, Web AR in school instructional material to keep the teaching methods in line with the tremendous technological progress and invest students' knowledge in this field. The main objective of the current study is to develop a Web AR application to investigate the effect of Web AR on the achievement of 4th year primary school pupils in English. The pupils at this level focus on moving and interacting [6]. The currently applied classroom techniques may not give those teachers sufficient opportunities to explore a new method for teaching EFL for pupils. The procedure of Web AR application includes two basic steps. Firstly, selecting the convincing Web AR development platform according to three evaluation criteria. Secondly, developing the Web AR with 73 English vocabularies included in the last four units of the English pupil's book (4th level). The procedures of the study include steps. First, select randomly a sample of pupils and assign them to the Experimental Group (EG) and Control Group (CG). Second, equalize the elected pupils in the factors that may affect their performance. These factors include, students' age in months, parents' academic attainment, and their scores in the pre-test. Then, EG and CG have been taught English for 12 weeks, and finally, three achievement posttests are constructed in order to assess the pupils' performance.

The following hypotheses are supposed to be verified in order to attain the desired goals:

- In the first post test, there is no discernible difference between the CG achievement's and the EG achievement's mean scores.
- The second post test's mean scores for the CG achievement and the EG achievement do not significantly differ from one another.
- In the third post test, there is no discernible difference between the CG achievement's and the EG achievement's mean scores.

These hypotheses are established because the students are randomly selected to either group, we can presume that the groups are fundamentally identical in comprehension capabilities. Furthermore, the experimental environment is identical, and the experimenters conducted the same conditions and procedures with both groups (including same teachers for both groups). Hence, researchers can confidently conclude that any variations between groups are due to the variance in the appliance.

This study and the developed Web AR application present the following contributions:

- Developing Web AR application with 73 English vocabularies included in the last four units of the English pupil's book (4th level).
- The conducted experiments teaching EFL to both control and EG last for one semester (12 weeks).
- Adopting three exams to obtain accurate results from the conducted study.
- Conducting three evaluation criteria on two Web AR development environments (sample one and sample two) to select the convincing Web AR development for this study.

This study is structured as follows: Section two discusses the applications of AR used in the context of English learning, MAR limitation and review of the Web AR technologies. Section three shades the light on Setting up the convening Web AR infrastructure. The proposed Web AR application and the design of the experiment are presented in Section four. In Section five, the analysis and the results of the experiment are presented. Lastly, the conclusion is in section six.

## 2. Background and Related Works

The previous studies stated in the following Sections 2.1 and 2.2 are gathered via a systematic search. Initially, generic terms are used in order to ensure that the writers have covered most of the related research papers. "Augmented reality" and "primary school" or "elementary school" or "grade school" are the key search expressions used in this study. In order to collect high-quality data, the writers have searched two popular databases (IEEE Xplore and Science Direct) refined according to the last 6 years. The search results are carefully checked based on the papers' titles, summaries, keywords, and conclusions. Inclusion and exclusion criteria are finally applied to 5 research papers related to English with AR and more than other 20 research papers related to other primary school subjects with AR. The last part (Section 2.3) reflects the writers attempt to explore ways to take a step forward in the field of Web AR development and deployment.

### 2.1. English in Augmented Reality Applications

The following recent prominent studies on AR and English primary education were collected according to a systematic search.

The study on [7] aimed to help third-grade primary school students to spell and learn English vocabulary. It was designed to answer the question, "Do learning styles matter?" It aimed to investigate the effectiveness of students with different learning styles. Two teaching methods were used, the self-directed learning approach, where the teaching did not restrict the learning sequence, and the task-based learning approach, which limits the research questions. The study concluded that both methods had similar and high learning effectiveness. The study did not mention the period of the experiment, the number of vocabularies used in the study, the operating system used, and the sample of the EG was limited.

The study on [8] aimed to investigate the effectiveness of English as a Second Language (ESL) with a mobile-based AR application. Face-to-face interviews with the teachers and students show that AR in ESL has some unique advantages. The interview was adopted to collect the require data, which was considered an inaccurate way, especially for primary school students.

The researchers of [9] developed a digital game-based learning system based on AR, and selected elementary English letters and common vocabulary as the teaching materials to develop a game system for English learning. The study aimed to improve motivation and effectiveness of learning English vocabulary. The researchers did not mention the experiment period or the number of vocabularies used in the study. The study concluded that digital game-based learning can improve the motivation of learning English vocabulary.

The writers on [10] explored children's experience in terms of knowledge gain and enjoyment when learning through a combination of AR and speech recognition technologies. The proposed teaching strategy was effective in teaching colors and shapes. The results were satisfying, according to what the writers concluded. The writers on [11] investigated

the effect of multi-sensory AR on students' motivation in English language learning. The period of the study was 10 weeks. The writers concluded that the combination of multi-sensory and AR is effective and can motivate students for learning English.

However, the period of their studies is less than one month. Except for [11] study that last for 10 weeks. Whereas, it is constant in English teaching methods that pupils' psychological, social, and emotional aspects are not affected by a short duration [12], especially the tendency and direction [13]. Moreover, all previous studies relied on the pretest to ensure that the pre-performance of the EG and CG is equal. Furthermore, they have not equalized the involved sample in other factors, such their age and their parents' academic attainment. The current study is conducted in Iraq, which is considered a developing country. Thus, the investigation of the parents' academic attainment is essential. Home-based parental involvement can affect children's performance and engagement at school [14,15]. Previous studies have not adequately describe the instructional material or the number of AR the English vocabulary conducted in the experiment. Also, they have not explain the traditional method used for learning EFL or ESL.

### 2.2. Limitations and Challenges of Mar Application in Primary School

The previous studies were mostly not sufficiently clear or thorough in describing the methods, procedures, and limitations, which were consistent with what was mentioned in [16]. What will be presented here may be considered as clear challenges, limitations, or restrictions in using MAR. The writers argue that most of these restrictions and challenges that will be mentioned and discussed can be decrease or eliminated by shifting from MAR to Web AR technology.

*Lack of privacy:* What drew the researchers' attention is that all MAR applications that were used in the previous studies were not published in authorized and recognized stores (Google Play, Apple Store, APKPure, etc.). Here came the legitimate parents' fears of installing a strange application on their devices that was not reviewed and published officially [17]. Furthermore, no mention was made of the method to check the reliability of the MAR that was used except in the study conducted by [18,19]. The usage of MAR development libraries also created the potential for a hidden operations danger, where a dishonest or inexperienced application developer might do additional vision operations in the background of an otherwise honest AR application without informing the end user [20].

*Lack of Scalability:* The process of modifying and adding components in the MAR application was almost a difficult task, especially if the application is not published in an official mobile applications store. This is not the case in Web AR, as it is a one-time modification in the web server. The limitation of many studies is the application's need for more 3D objects [21] or the need for editing the contents [22] or sometimes the platform does not accommodate the learning requirements of different students [23,24], or the need to create a new feature like feedback as in [25,26].

*Lack of access:* Considering that Bring Your Own Device (BYOD) policy is not yet implemented in most primary schools. Students can not try MAR application they applied during the experiment at home [27], taking into consideration that occasionally, the time for the experiment was too short, so the students could not remember the content that had just been taught [28]. Even more, sometimes, the experiment classroom equipment that must be available to complete the experiment was not enough for all students [29], which in turn can lead to a low number of participants as in the study [30]. Furthermore, some students in [31] complained that they experienced neck stiffness because they looked at the screen of the tablet during the experiment for too long time.

*Complex MAR:* teachers complain about complex configuration and utilization [30,32]. Rarely, they cannot complete the experiment without technical support [18,33,34]. Meanwhile, complex applications can increase students' cognitive load [32]. Rationality, the authors of [35] suggested that both teachers and students need to be trained in order to be exposed to the use of AR.

*Platform dependency:* most of the MAR applications that were developed support Android devices [18,30,33–37], although the rest of the research papers did not mention the utilize operating system. However, we believe most are using Android development environments since Android devices are cheap and affordable. The vast majority of the developed MAR applications were developed using Vuforia and unity Software Development Kit (SDK), which in the end supports several devices [38].

### 2.3. Review of Web Ar Technologies

There is a trend that AR needs to meet the web. The reason is most mobile AR applications are built for a specific platform with a lack of cross-platform support. To reach all users, an AR developer needs to repeat the development cycle to accommodate different platforms [39] the problem does not stop there, the well-known mobile AR SKD normally supports a specific list of devices [40,41].

According to Cisco's 2020 Global Networking Trends Report, AR and VR traffic will increase and multiply by the end of 2022, and by the end of 2025, cloud-based AR applications will be the first new network business request [41]. The massive development of mobile computing and emerge a new direction for mobile AR on the web. The solutions contributed to the implementation of Web AR can be divided into Java-based and Browser-based solutions [42]. Table 1 summarizes Web AR solutions and enabling technologies.

**Table 1.** Webar Solution and Enabling Technologies.

| API Tech. Lib. | Short Description |
| --- | --- |
| Awe.js [43] | JavaScript-based API released in 2012, relies on Jthree.js, JSARToolkit, ARToolkit [44]. Hello world time required less than 10 min [45]. Support marked-based AR, all standard browsers and can operate with desktop PC. |
| A-Frame AR | JavaScript-based API released in 2016, relies on A-Frame, three.ar.js, WebARonARKit/Core, and WebXR. Hello world time require less than 30 min [46]. Does not support marked-base AR and desktop PC. Does not operate on all web browsers. |
| Ar.js [42] | JavaScript-based API released in 2017, relies on A-Frame, three.js, and ARToolKit. Hello world time require less than 10 min [47]. Support marked-based AR, all standard browsers and can operate with desktop PC. |
| mind-AR-JS [48] | JavaScript-based API released in 2022, relies on A-Frame, three.js, and ARToolKit. It supports image tracking and face tracking. Support marked-based AR, all standard browsers and can operate with desktop PC. |
| Wikitude [49] | A browser kernel-based solution for Web AR was released in 2008. Can provide the experience of Location-based AR, image recognition and tracking, and geolocation technology. Can operate on Android, IOS, and Windows. |
| BlippAR [50] | A browser kernel-based solution for Web AR was released in 2011. Can provide the experience of image recognition and tracking, and support Google glasses. Can operate on Android and IOS. |
| Argon4 [51] | A browser kernel-based solution for Web AR was released in 2016. Can provide the experience of any 3D view of reality to be augmented and viewing many WebAR at the same time. Can operate on Android and IOS. |
| Three.js [52] | A cross-browser JavaScript framework and application programming interface (API) called Three.js was launched in 2010 and is used to produce and display animated 3D computer graphics in a web browser using WebGL. Three is referred to as 3D animation. |
| WebGL | WebGL (Web Graphic Library) it is a Java-based API cross-platform, royalty-free API released in 2011, used to create 2D and 3D graphics in a Web browser [52] this provides a hardware-based (Graphic Processing Unit) rendering acceleration approach on the Web [53]. |
| WebRTC | WebRTC (Web Real-Time Communications) Free open source project comprises a set of technologies and standards that provide real-time communication with web browsers released in 2011 [54]. It allows audio and video communication to work inside web pages by allowing direct peer-to-peer communication, eliminating the need to install plugins or download native apps [55]. |
| a-frame | Open-source web entity component system framework for Three.js where developers can create 3D and Web VR scenes using HTML released in 2015 [56]. In the middle of 2015, the Mozilla VR team created A-Frame. A-Frame is a Web VR framework that speeds up and simplifies the creation of virtual reality experiences by allowing HTML coding instead of the robust but difficult WebGL. |
| WebXR | API that supports devices for accessing and presenting AR and VR on the web browser. released in 2018 [57]. |

In addition to the above Web AR solutions, many online platforms with in-house cloud-based technologies enable the user to create, automate and manage Web AR applications.

These cloud-based Web AR platforms empowered off-computation fetcher to accelerate the computing process [55]. In this study, mywebar platform [58] adopted for the reasons and justifications that will be discussed in the following section.

## 3. Setting up the Web AR Infrastructure

There are many factors that may influence what technologies/tools the researchers select to deliver effective and accepted E-learning experience. These factors may include study objectives, learner access to the technology, tool complexity, comfort and familiarity with technology and easy to access [39].

The compulsory transition of students to remote online education during the COVID-19 pandemic encouraged the choice of using Web-based AR. For about two years, pupils interact daily with the internet (Google Classroom and WhatsApp in the schools where the experiment is held) to accomplish their school. Those pupils have faced a lot of technical difficulties. Thus, the authors do not want to repeat the bad experience with performance decrease, network instability, and slow internet speed [6]. Therefore, choosing the appropriate educational tool that ensures stability and reliability is essential in this study. In this context, three evaluation criteria are carried out for choose this study Web AR development environment. The three evaluation criteria have been carried out on two Web AR samples. Sample one, built using the three.js library and mindAR library. three.js is a JavaScript library dedicated to designing and depicting animated 3D models and computer graphics displayed on a browser using WebGL. MindAR is an open-source web AR library integrated with the three.js library [59]. The mindAR provides image tracking capabilities with an online preprocessing step that compiles the image. After building the Web AR application and successfully run it locally. Sample one uploaded to GitHub page hosting service [60]. Sample two developed using mywebar cloud-base development environment. The result of this phase is described in the Figure 1 below.

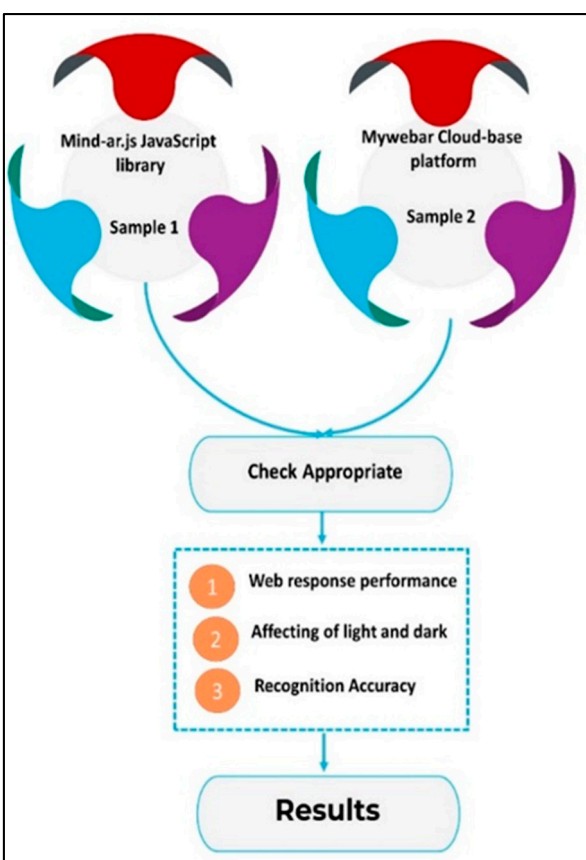

**Figure 1.** Flow of Web AR samples test.

### 3.1. Web Response Performance Test

The Website's performance measures how quickly a website's pages load and display in the client's web browser [61]. The selection of designing, and developing website tools are essential to delivering an effective and high-performance website [62]. Bad performance can decrease user interaction with the website and reflect negative feedback. In this study, the performance of both ar.js and mywebar Web AR samples measure using the GTmetrix [63] web application. GTmetrix is an accurate [64] and free web application to analyze the page speed and rate the score of the performance and structure based on more than 15 metrics. GTmetrix results favored sample 2 with a performance rate of 85% and 68% for sample 1.

### 3.2. The Affecting of Light and Dark

AR applications designed as marker-based, can be affected technically by light or dark to recognize the marker [29]. Lighting Minimize possible recognition errors [26]. Consequently, the authors conducted a set of experiments to measure and test the performance of Web AR samples with respect to different lighting settings. Table 2 shows the results of ar.js and mywebar Web AR samples where different lighting intensities are measured, ranging from 2445 to 12 luminous Flux Per unit area (Lux). The results show that the performance of sample two is better than that of sample one.

**Table 2.** Various Lighting Test Samples Results.

| Lighting Samples | 2445 | 1328 | 843 | 431 | 380 | 147 | 35 | <12 |
|---|---|---|---|---|---|---|---|---|
| Sample One | Yes | Yes | Yes | Yes | Yes | NO | NO | NO |
| Sample Two | Yes | Yes | Yes | Yes | Yes | Yes | Yes | NO |

### 3.3. Recognition Accuracy

Some problems have been reported in studies that relied on Web AR, where "sometimes the application has problems detecting the vocabulary card" [6] to measure the recognition performance, both Web AR samples are tested separately by scanning the QR code for ten iterations. Accordingly, the Web AR accuracy is measured by computing the recognition accuracy of each QR code [64] using Equation (1):

$$\text{Accuracy \%} \frac{\text{QR recognized}}{\text{no. of attempts}} \times 100 \tag{1}$$

The obtained result show that the recognition accuracy of sample two and sample one are 80% and 60%, respectively.

The three evaluations criteria results favored sample 2, which will be adopted in this study. Mywebar is a cloud-based Web AR development environment within house computer vision technology [65] developed by DEVAR entertainment, which is a technology company that specializes in the development of augmented reality content and products [66].

## 4. Methodology

This section sheds light on: the Web AR application development and gives an overview of mywebar development process. Following, the procedures of the study include steps. First, select randomly two groups of pupils and assign them to EG and CG. Second, equalize the elected pupils in the factors that may affect their performance. These factors include, students' age in months, parents' academic attainment, and their scores in the pre-test. Then, the EG and CG have been taught English for 12 weeks. Finally, three achievement posttests are constructed in order to assess the pupils' performance.

*4.1. The Web AR Application Development and User Flow*

The Web AR development starts once you select the mywebar development plane and create new project. Mywebar dashboard is the main Web based interface where Web AR projects can be managed and different features can be set. Many multimedia sources can be added with a huge number of extensions. After inserting the vocabulary 3D models (or image), sound file, adding text and setting up the Web AR seen, the user can quickly scan the QR code and browse the custom domain to enjoy the Web AR experience. Figure 2 shows the Web AR development overview.

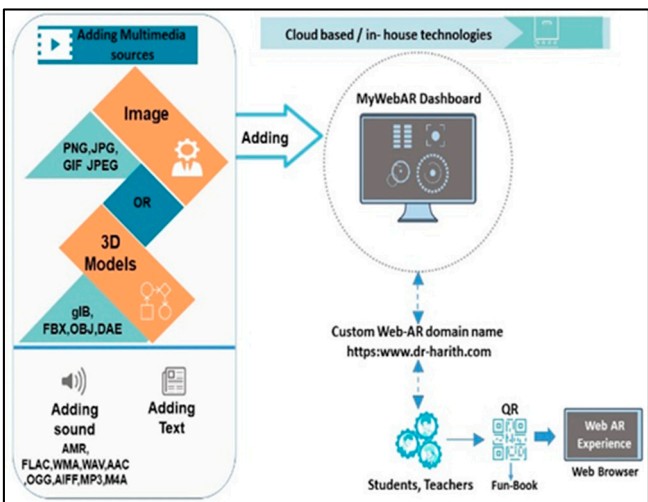

**Figure 2.** Web AR Development Over View.

Mywebar support adding many objects, including images, video, 3D models, audio, text, light source, shapes, cylinder video, and JavaScript-based custom plugins. Below is a detailed explanation of these objects.

Adding image: It can be uploaded or selected from previously uploaded files. The position of the selected image over the QR code can be easily adjusted via a computer mouse. For adding images, mywebar supports PNG, JPG, GIF, and JPEG extensions. For image tracking, it is recommended to use high-contrast images with no repeating patterns or shapes and contain a lot of detail.

Adding 3Dmodels: mywebar support a wide range of 3D model extensions including, GLB, FBX, OBJ, and DAE. 3D models can be imported directly from pc or by linking the Sketchfab [67] platform account to mywebar. The tools board can be used on the 3D Workspace to change between rotating, moving, and changing the size of the 3D models. It is discernible that 3D models in a web browser may work slightly differently than the software specially designed for them. That is why checking the models for errors is essential before uploading. It is recommended to use babylonjs [68] for testing 3D models. This site lets the developer check for missteps and deal with them.

Adding video: mywebar supports a wide range of video extensions including, 3GP, FLV, AVI, MKV, M4V, MP4, MOV, MPEG MPG, WEBM MTS, VOB and WMV. There are many ways to start video playback. We can auto-play the video once the scene is loaded, and play the video by tapping on it and by tapping on a virtual button.

Adding sound: mywebar also supports a wide range of sound extensions including, AMR, FLAC, WMA, WAV, AAC, OGG, AIFF, MP3, M4A. Sounds can be triggered on a button action and can be repeated many times.

Button on the action: Mywebar allows developers to assign many interactive events to any object on the scene. These events include playing audio on click, opening a webpage on click, playing 3D model animation on load, playing 3D model animation on click playing our pause embedded media on click, move, moving object to a new position on click, moving object to position on load and showing our hiding object on click.

Custom WebAR Domain: MyWebAR allows developers to use their domain name. The URL or domain name is what visitors type into their web browser or scan the website QR code to enjoy the web-based augmented reality experience. Firstly, the visitor requests the customized domain name; then, the request will be directed to the mywebar Webserver. Mywebar generates HTML code for developers, giving them additional flexibility, allowing them to incorporate augmented reality experiences with other material into already-existing websites using an <iframe> tag. The mywebar Webserver automatically provides the SSL (Secure Sockets Layer), but the developers can use their SSL.

Custom three.js JavaScript: Mywebar allows developers to integrate ready-made JavaScript code in the platform's advance code editor. Moreover, developers can import JavaScript-based custom plugins. The user flow of the proposed Web AR application includes three basic steps. First, students must point their mobile phone camera to the Web AR application QR code to browse and access the proposed application. Second, students are asked for their permission to access the device camera. Finally, students can scan the QR codes. Figure 3 shows the user experience of the proposed Web AR.

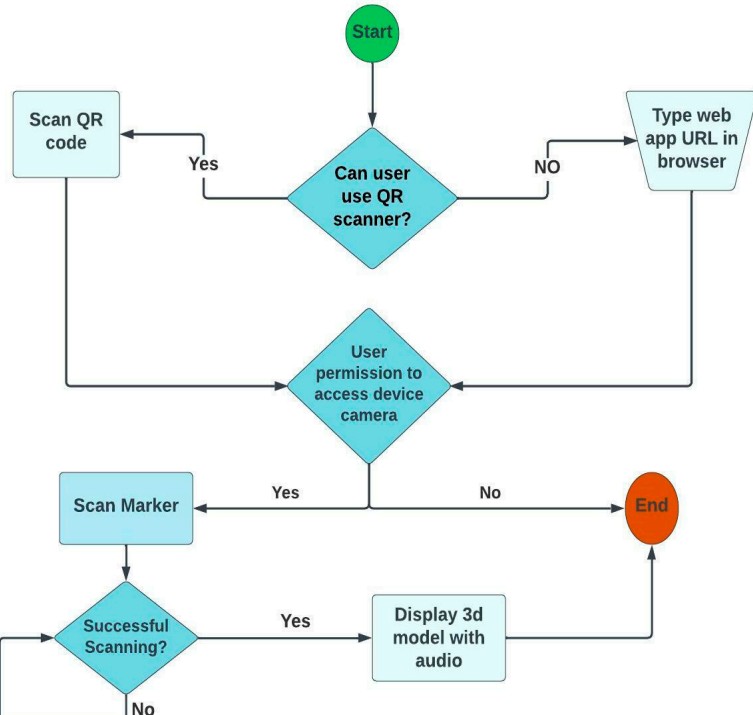

**Figure 3.** User flow through the proposed Web AR.

### 4.2. Experimental Design

An experimental design means testing hypotheses by a set of procedures [69]. "It is the blueprint of the procedures enabling the researcher to test the stated hypotheses and reaching valid conclusions about the relationships between independent and dependent variables" [70]. According to the following criteria, the experimental strategy used is known as the "Posttest Only Equivalent- Group Design":

1. Randomly choosing two student groups and assigning them to EG and CG.
2. Equalize the students in the two groups that are involved.
3. Administering just the EG and the independent variable.
4. The same lesson plans that were taught to the EG were used to teach the CG, albeit in a more conventional manner.
5. After, testing the two student groups involved.
6. Using the proper statistical tools to assess the data gathered and produce the desired results.

### 4.3. Sample Size and Population

The population of the current study includes 234 male and female pupils who represent all the fourth-year primary school pupils who are studying at ten coeducational private schools of the City of Tikrit/SalahDeen/Iraq in the academic year 2021–2022.

Al-Noor Coeducational Private School has been randomly selected to be the sample of the student. It includes 59 male and female pupils who are grouped into two sections, (A) and (B) and represent 25.21 of its original population. Section (A) whose number is 30 pupils has been randomly selected to be the EG and section (B) whose number is 28 pupils represents the CG, as shown in Table 3.

**Table 3.** The Population and Sample of Study.

| No. of Population | | Group | No. of Sample | | Total |
|---|---|---|---|---|---|
| **Male** | **Female** | | **Male** | **Female** | |
| 142 | 92 | EG | 14 | 16 | 30 |
| Total | | CG | 18 | 10 | 28 |
| 234 | | Total | 32 | 26 | 58 |

The Experimental Design include steps. First, select randomly a sample of pupils and assign them to EG and CG. Second, equalize the elected pupils in the factors that may affect their performance. These factors include, students' age in months, parents' academic attainment, and their scores in the pre-test. Then, EG and CG have been taught English for 12 weeks, and finally, three achievement posttests are constructed in order to assess the pupils' performance. as shown in Figure 4:

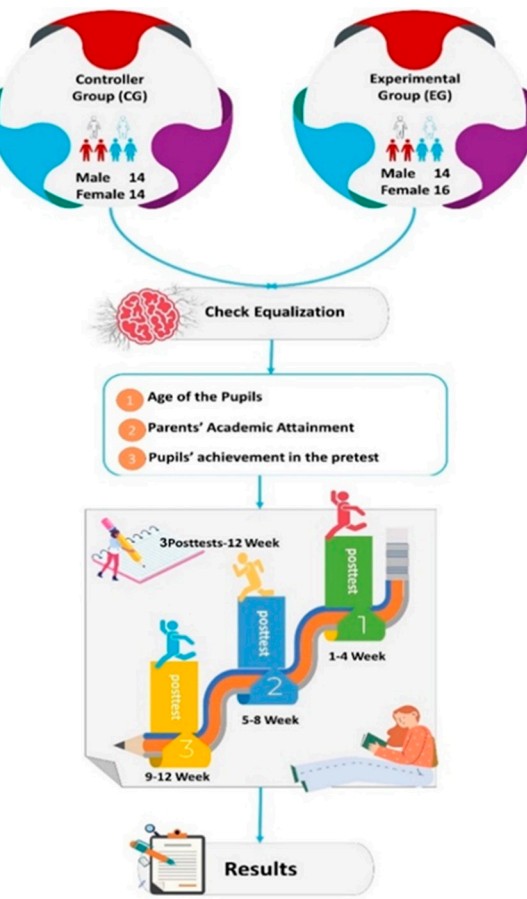

**Figure 4.** Experimental Design Overview.

### 4.3.1. Age of the Pupils

The pupils' age in months for the two groups have been counted till the first day of March, 2022. The t-test formula for two independent groups was used, it is found that there is no significant difference between the pupils of the two groups in their age, since the calculated t-value 0.38 is less than the tabulated t-value 2000, at a level of significance (0.05) and a degree of freedom (58), as shown in Table 4.

**Table 4.** The Mean Score, Standard Deviation, And T-Values Of Pupils' Age In Months.

| Group | No. of Subjects | Mean ×2 | S.D. | T-Values | | D.F. | Level of Significance |
|---|---|---|---|---|---|---|---|
| | | | | Computed | Computed | | |
| EG | 30 | 112.0333 | 2.88257 | 0.38 | 2.00 | 58 | 0.05 |
| CG | 28 | 111.7500 | 2.77055 | | | | |

### 4.3.2. Parents' Academic Attainment

a.    Fathers Academic Attainment

To ascertain whether there is a major disparity in the academic performance of the students' fathers, Chi-Square formula is utilized. Results indicate that the calculated value is 0.06, which is less than the tabulated value 5.99, at a degree of freedom (58) and a level of significance (0.05), as shown in Table 5. This means that there is no significant difference between the two groups of pupils in their fathers' academic attainment.

**Table 5.** Frequency And Chi-Square Value For The Fathers' Academic Attainment Of Both Groups.

| Stages of Education | Group | | Total | Degree of Freedom | Level of Significant | Chi-Square Value | |
|---|---|---|---|---|---|---|---|
| | E | C | | | | Calculated Value | Chi-Square Distribution |
| illiterate & Primary | 7 | 9 | 16 | 2 | 0.05 | 0.60 | 5.99 |
| Secondary &Diploma | 9 | 8 | 17 | | | | |
| bachelor | 14 | 11 | 25 | | | | |
| total | 30 | 29 | 59 | | | | |

b.    Mothers' Academic Attainment

In order to determine whether the academic achievement of the children' mothers differs significantly from one another, Chi-Square formula is utilized. Results indicate that the calculated value is 0.26, which is less than the tabulated value 5.99, Table 6 displays the results at a threshold of significance of 0.05 and a degree of freedom of 57. This indicates that there is no discernible difference in the academic standing of the mothers of the two groups of students.

**Table 6.** Frequency and Chi-Square Value for the Level of Mothers Academic Attainment of Both Groups.

| Stages of Education | Group | | Total | Degree of Freedom | Level of Significant | Chi-Square Value | |
|---|---|---|---|---|---|---|---|
| | E | C | | | | Calculated Value | Chi-Square Distribution |
| illiterate & Primary | 12 | 12 | 24 | 0.05 | 2 | 0.26 | 5.99 |
| Secondary &Diploma | 10 | 9 | 19 | | | | |
| bachelor | 8 | 7 | 15 | | | | |
| total | 30 | 28 | 59 | | | | |

c.    Pupils' Achievement in the Pretest

The two groups of pupils have been subjected to the pretest which is constructed in terms of the first three units of the pupils' book (English for Iraq, 4th primary). Then, testees' responses on the pretest have been calculated and analyzed by using the t-test formula for two independent samples. Results show that there is no significant difference between the mean scores of the two groups' achievement in the pretest, at (0.05) level of significant and (58) degree of freedom since the computed t-value 0.21 is less than the tabulated t-value 2.01, as shown in Table 7.

**Table 7.** The Mean, Standard Deviation, and T-Value of Pupils' Scores in the Pretest.

| Group | NO. of Pupils | ×2 | SD | T-Value | | D.F | Level of Significant |
|-------|---------------|-----|-----|---------|----------|------|---------------------|
| | | | | Computed | Tabulated | | |
| EG | 30 | 71.60 | 15.12 | 0.21 | 2.01 | 58 | 0.05 |
| CG | 28 | 70.44 | 13.72 | | | | |

d.    Teachers Training on Web AR Application

Two teachers at Al-Noor Coeducational Private Primary School have been trained on the process of the Web AR Application for a period of five days. Those two teachers are required to teach the EG of pupils, in terms of the steps of the Web AR application. The first teacher has got a Bachelor degree in English. She has six-year experience in teaching EFL. The second teacher has got a Bachelor degree in Computer Sciences. She has a four-year experience in the field of Information Technology (IT).

e.    Instructional Material and Pupils' Instruction

The instructional material includes the last four units of the English Pupil's Book, English for Iraq (4th primary) namely: Unit five, six, seven and eight as well as a specific manual. These units are taught to the two groups of pupils for a period of 12 weeks, at a rate of four lessons a week during the second course of the academic year 2021–2022 it is worth pointing out that the pupils are encouraged to use their manuals at home. Unit five, six and seven includes a topic linked story with colorful illustrations, narrative text and speech bubbles that help contextualize learning and support reading, while unit eight is a revision unit.

The EG is taught inside the computer laboratory of the school according to the steps of the Web AR technology, whereas, the CG is taught according to steps of the traditional method, i.e., without the Web AR technology, as follows:

1.    Lesson Plan for Teaching the EG

Class and Section: 4th (A)
Date: 2 March 2022
Time: 8:00–8:40
Unit: Five
Topic: Places to visit
Aim: Learn and read for talking about places to see.
The devices used during the given lessons are shown in Table 8.

**Table 8.** Devices Used in the EG.

| NO. | Device | Model |
|-----|--------|-------|
| 1. | Promethean interactive whiteboard | PRM-AB378-02 |
| 2. | Vivitek Projector and Mount | DH758UST |
| 3. | Promethean Active Soundbar | |
| 4. | Huawei Tablet | |

Steps of presentation: the EG has been taught the concept of Web AR and join it with the instructional games that are given in the pupil's book, such as Pokémon game and some of the AR applications that are existed in the mobile phones of the pupils. Teachers use the interactive display system to explain the function of Web AR application on some of the English vocabulary of the given lesson show the pupils those vocabularies in 3D module, while they are listening to the recorded sound.

This step is accomplished by using the audio system which is blended with the smart board of the computer laboratory.

Each pupil is given a Brochure, which is called a fun-book. It contains a few presented vocabularies. The name of each pupil as well as the QR of Web AR address are stuck on the fun-book, as shown in Figure 5.

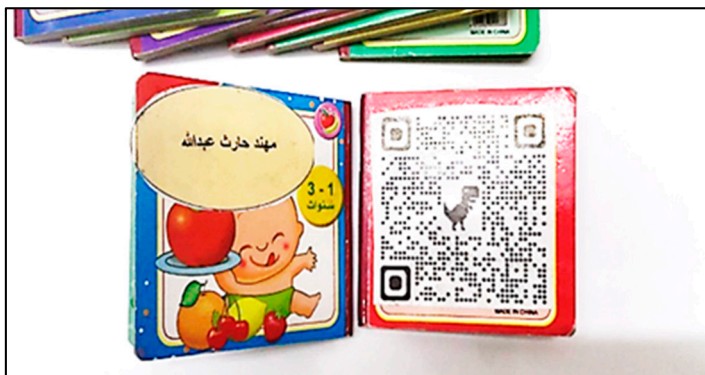

**Figure 5.** Fun-Book Brochure.

The involved pupils are reorganized into teams, small groups or pairs in order to try scanning QR codes by themselves under their teachers' supervision. In the second week of teaching by using the Web AR technology, each pupil is given a manual that contains the vocabularies of the three intended units and explains clearly of how to use the Web AR application. the pupils can use their manuals at home in order to train themselves on learning English through the Web AR application by using any available mobile device at home.

2.    Lesson Plan for Teaching The CG

Class and Section: 4th (B)
Date: 2 March 2022
Time: 8:00–8:40
Unit: Five
Topic: Places to visit
Aim: learn and read words for talking about places to see
Steps of presentation: the CG has been taught
EFL as follows:

- Discuss the given pictures and establish the context read the rubric and discuss it.
- Explain the task and use illustrations to support meaning
- Play the track song, pupils listen to the given words while following in their books.
- Play the track a second time; pupils listen and point to the appropriate pictures.
- The pupils listen and read aloud the individual words, phrases and short text. Reading aloud reinforces sound/spelling relationship.
- Organize the pupils into teams, small groups or pairs by asking them to work with pupils who sit near them.
- Ask the pupils to match the presented words to the given pictures and definitions.
- Ask questions and elicit answers from the pupils. Play track again to confirm answers if necessary.

f.    Construction of the Posttest and the Scoring Scheme

An achievement posttest is created based on the behavioral and content objectives of the instructional materials that were presented. There are five questions and 13 items on each test. and its total scores are 100, i.e., twenty marks are specified for each question, as shown in Table 9.

**Table 9.** The Specifications of the Contents, Behaviors, and Scores of the Three Posttest.

| No. of Question | Content | Behavioral Objectives | No. of Item | Score |
|:---:|:---:|:---:|:---:|:---:|
| 1 | Unseen Passage | To write (Yes) for the correct item and (No) for wrong item | 2 | 20 |
| 2 | Tense | To fill in the blanks with the option of the appropriate tens | 4 | 20 |
| 3 | Making Question | To match between the list of questions and the list of answer | 2 | 20 |
| 4 | Vocabulary | To match between the given words and their pictures | 4 | 20 |
| 5 | Punctuation | To punctuate the given item | 1 | 20 |
| | Total | | 13 | 100 |

The first question includes an unseen passage with two true/false items. So, 10 marks are specified for each correct answer and zero for a wrong one. The second question includes four items and five marks are specified for each correct item and zero for a wrong one. Question three includes two items to match between the list of questions.

g.    Pilot Administration of the Posttest

The pilot administration of the three posttests has been utilized to test the required time for answering their questions and to know whether the questions are clear for the testees. Thus, each posttest has been administrated to the pilot sample of 15 pupils selected randomly from Ibn Al-Haytham Cooperative Private School. Results indicate that the given tests instructions are clear to the testees and the time needed to answer the questions of each test ranges between 30 to 40 min.

h.    Validity, Reliability, and Items Analysis

The constructed three posttests are validated as they have been constructed in terms of the specified content and behaviors. Then the face validity of each test is also ensured by exposing it to a jury of specialists in English methodology. Each test reliability is calculated by using Alpha Cronbach formula. The obtained degree of reliability ranges between 0.84 and 0.88 for the three posttests.

Item difficulty level reflects the percentage of testees who answer the item properly. The difficulty level of the most appropriate test items ranges from 0.15 to 0.85 [3]. The obtained difficulty level of the constructed items ranges between 0.35 to 0.75, as shown in Table 10.

Item discrimination power refers to the degree to which an item distinguishes between good and poor testers. According to Ebel [71] the discrimination power of each item should be within the average of 0.30 to 0.70 in order to be acceptable. The obtained discrimination power of the intended items ranges between 0.30 to 0.60, as shown in Table 10.

i.    Posttests' Final Administration

The first exam was administered to both sets of students on 1 June 2022, after the three posttest items' validity, reliability, degree of difficulty, and discrimination power were confirmed. The concerned test-ers have received their second test papers, and they are now expected to carefully read the provided instructions and clearly write down their responses on their test paper within the allotted 37 min. The test papers have finally been gathered and will be scored.

**Table 10.** The Difficulty Level and Discrimination Power of Posttest Items.

| Question | Item/S | High | Low | DL | DP |
|---|---|---|---|---|---|
| 1 | 1 | 8 | 5 | 0.3 | 0.65 |
|  | 2 | 7 | 4 | 0.3 | 0.55 |
| 2 | 1a | 8 | 3 | 0.5 | 0.55 |
|  | 2a | 9 | 6 | 0.3 | 0.75 |
|  | 1b | 8 | 5 | 0.3 | 0.65 |
|  | 2b | 8 | 4 | 0.4 | 0.6 |
|  | 3b | 8 | 5 | 0.3 | 0.65 |
| 3 | 1 | 10 | 4 | 0.4 | 0.35 |
|  | 2 | 9 | 5 | 0.4 | 0.7 |
|  | 3 | 8 | 4 | 0.4 | 0.6 |
| 4 | 1 | 9 | 3 | 0.6 | 0.6 |
|  | 2 | 10 | 4 | 0.6 | 0.7 |

## 5. Analysis of Data and Discussion of Results

### 5.1. Comparison between the Achievement of The EG and CG in The First Posttest

The mean scores as well as the standard deviations of the two groups have been acquired in order to confirm the first hypothesis, which claims that there is no statistically significant difference between the mean scores of the EG achievements and that of the CG achievements in the first posttest. According to the results, the CG's mean score is 9.18, while the EG's is 9.13. In order to determine whether there is a statistically significant difference between the acquired mean scores, the t-test formula for two independent samples is utilized. According to Table 11, the computed t-value is 0.14, but the tabulated t-value is 2.00 at 56 degrees of freedom and 0.05 level of significance. This outcome indicates that there.

**Table 11.** The Mean Scores, Standard Deviations, and T-Values of the EG and CG in the First Posttest.

| Group | No. of Pupils | Mean Scores | SD | T-Value | | DF | Level of Significance |
|---|---|---|---|---|---|---|---|
| | | | | Computed | Tabulated | | |
| EG | 30 | 9.133 | 1.479 | 0.14 | 2 | 56 | 0.05 |
| CG | 28 | 9.18 | 0.98 | | | | |

### 5.2. Comparison between the Achievement of the EG and CG in the Second Posttest

In order to verify the second hypothesis which states that there is no significant difference between the mean scores of the EG achievement and that of the CG achievement in the second posttest, the mean scores as well as the standard deviations of the two groups are obtained. Results show that the mean scores of the EG and CG achievements are 9.67 and 9.47, respectively. In order to determine whether there is a statistically significant difference between the acquired mean scores, the t-test formula for two independent samples is utilized. At 56 degrees of freedom and 0.05 level of significance, the computed t-value is determined to be 1.02, while the tabular t-value is 2.00, as shown in Table 12. These results indicate that there is a little improvement in the EG performance in the second posttest. It is worth pointing out that the obtained improvement is not significant thus, the second hypothesis is also rejected.

**Table 12.** The Mean Scores, Standard Deviations, and T-Values of the EG and CG in the Second Posttest.

| Group | No. of Pupils | Mean Scores | SD | T-Value | | DF | Level of Significance |
| | | | | Computed | Tabulated | | |
|---|---|---|---|---|---|---|---|
| EG | 30 | 9.67 | 0.71 | | | | |
| CG | 28 | 9.47 | 0.79 | 1.025 | 2 | 56 | 0.05 |

*5.3. Comparison between the Achievement of the EG and CG Achievement in the Third Posttest*

In order to verify the third hypothesis which states that there is no significant difference between the mean scores of the EG achievement and that of the CG achievement in the third posttest, the mean scores as well as the standard deviations of the two groups are obtained.

The results indicate that the EG and CG achievement mean scores are, respectively, 9.97 and 9.61. In order to determine whether there is a statistically significant difference between the acquired mean scores, the t-test formula for two independent samples is utilized. According to Table 13, the computed t-value is determined to be 3.00, whereas the tabulated one is 2.00, at a degree of al freedom of 56 and a level of significance of 0.05.

**Table 13.** The Mean Scores, Standard Deviations, and T-Values of the EG and CG in the Third Posttest.

| Group | No. of Pupils | Mean Scores | SD | T-Value | | DF | Level of Significance |
| | | | | Computed | Tabulated | | |
|---|---|---|---|---|---|---|---|
| EG | 30 | 9.97 | 0.18 | | | | |
| CG | 28 | 9.6 | 0.62 | 3 | 2 | 56 | 0.05 |

The obtained results indicate that there is a significant difference between the achievement of the two involved groups, and in favors of the EG. Hence, the third hypothesis is rejected.

Results of the current study have proved the effectiveness of the Web AR technology for learning EFL to primary school pupils. They have also proved the necessity of having more practice in using the intended technology for developing pupils' performance. The positive role of the Web AR technology could be attributed to a variety of factors, as follows:

1. The Web AR technology enables the pupils to be involved using the lesson material communicatively.
2. It motivates the pupils and encourages them to participate in the classroom activities.
3. Having a long period of learning on using the Web AR is significant, as the pupils tendencies and acceptance of the experiment tools require time and effort.
4. Teaching EFL by using Web AR technology is enjoyable and desirable. The involved pupils show their interest and enthusiasm to learn the presented material.
5. The Web AR technology could be used easily by EFL teachers inside their classrooms to facilitate the process of teaching English material to their pupils. There is no complain about technical problem since, no installation and updating is required.
6. The QR code scanning can be a useful, interesting, and entertaining learning tool.
7. The pupils could also use the Web AR technology at home to enable them learning English material.

The limitations of the current study include the lack of previous research studies on the topic of Web AR technology, which result no comparisons with previous research. The other limitation of this study is that our experiments' results cannot be generalized because the tests are carried out only on coeducational private school where the available resources and tools contributed to the experiment success. The pupils did not register any complaints about the poor internet service in their school, since the school is equipped with a fiber internet connection. Complaints are recorded about the response of the Web AR application due to the slow internet service in pupils' home, especially during the bottleneck time from 9:00 to 11:00 p.m.

## 6. Conclusions

Primary school pupils should be supplied with new and interesting technology that could be used easily for learning English inside their classrooms and their home. The result of this study augmented our aim to work on integrating Web AR technology in school instructional material to keep the teaching methods in line with the tremendous technological growth and invest students' knowledge in this field.

The long period of experiment study is significant, as the student's tendencies and acceptance of experiment tools require time and effort. In teaching English, Web AR technology helps pupils to improve their achievement and can create an enjoyable atmosphere and break classroom boredom. Using Web AR technology enables non-native pupils to listen to the pronunciation of the presented material and see it simultaneously. Moreover, having a lot of practice and opportunity in using the Web AR technology encourages pupils' brainstorming and arouses their interest by using words and imagination which eventually improve the pupil's performance in English.

Web AR technology can be considered as easy to access, easy to use, scalable, secure, and platform/device independent. Although, there are technical challenges or limitations related to the ease of Web AR deployment due to the restrictions of web browsers capability and 3G/4G wireless networks. New emerging 5 G and B5G networks and the development in the field of JavaScript-based API can partially decrease many of the limitations.

According to the obtained results and drawn conclusions, Iraqi Ministry of Education is recommended to supply the schools with the facilities and devices required for the employment of modern technology for teaching English to pupils. The directorates of education have to involve their EFL teachers into in service training workshops on the employment of the computer-aided technology inside their classroom. Primary school teachers are advised to use the Web AR technology in teaching EFL to enable their pupils learn the presented material easily.

For further studies, a question may be aroused as to whether the primary educational institution employs enough new technologies that qualify pupils to integrate with the labor market in the future. Future research should consider the potential effects of Web AR on other sciences. Moreover, it will be important that future research study pupils' Web AR access log understand their usage patterns better.

**Author Contributions:** Methodology, H.A.H.; Software, M.H.A. and Q.A.H.; Validation, M.A.-H.; Formal analysis, Q.A.H.; Investigation, R.D.I.; Resources, R.D.I.; Data curation, M.A.-H. and Q.A.H.; Writing—original draft, H.A.H. and M.H.A.; Writing—review & editing, H.A.H. and N.T.M.; Visualization, N.T.M.; Supervision, H.A.H. All authors have read and agreed to the published version of the manuscript.

**Funding:** This research received no external funding.

**Data Availability Statement:** All data were presented in the main text.

**Acknowledgments:** The authors would like to thank all the participants in the experiments, children, their teachers and management staff, in particular Rand M Mahir, Nuha Abdulla Rashid and Huda Hashim ghani from Al-Noor Coeducational Private Primary School, SalahDeen, Iraq.

**Conflicts of Interest:** The authors declare no conflict of interest.

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
