# Peer review of "The Effect of Web Augmented Reality on Primary Pupils’ Achievement in English"

_asi, doi:10.3390/asi6010018_

Round 1

Reviewer 1 Report

The research investigates the effect of Web Augmented Reality on Primary Pupils’ achievement in English. The reviewer has several questions that would appreciate the authors to make a little bit clear:

1.     What is the originality of the proposed research?

2.     Authors proposed several hypotheses but would be better to explain a little bit about how these hypotheses were developed.

3.     The reviewer would suggest to add a user flow:

https://www.productplan.com/glossary/user-flow/

4.     Is it feasible to have a look the prototype that authors created?

5.     Please check reference carefully. 

Reviewer 2 Report

Dear Authors,

Congratulations. It is a good paper. However, there are a few minor aspects to be fixed before the acceptance:

- There are many typo mistakes.

- In Table 10, it looks like the ITEM/S column is partially hidden.

- Some figures need to be explained a bit more, as for example, figure 3. 

Kind regards

Round 2

Reviewer 1 Report

Authors have addressed all my concerns.